# MGF-ESE: An Enhanced Semantic Extractor with Multi-Granularity Feature Fusion for Code Summarization

## Abstract

Code summarization aims to generate concise natural language descriptions of source code, helping developers to acquaint with software systems and reduce maintenance costs. Existing code summarization approaches widely employ attention mechanisms to assess the relevance between nodes in the Abstract Syntax Tree (AST), which generates context vectors that reflect the semantics of the source code. However, these approaches with AST fail to extract other granular features, such as code tokens and Control Flow Graph (CFG), which suffer from severe semantic gaps when capturing data and control dependencies. To address this issue, we design an enhanced semantic extractor with multi-granularity feature fusion (MGF-ESE) to improve the model capability in comprehending and processing the overall semantics of the code. Specifically, to process the AST more effectively, we present a novel AST generation method with compresses the scale of nodes to enhance the semantic information of each node. Then, we present a disentangled attention mechanism based on relative positional embeddings for further encoding. Moreover, we extract the code tokens and CFG of source code to supplement the syntactic and structural information, and further fuse them with the AST separately through cross-attention modules. Finally, extensive experiments on two public datasets show that MGF-ESE outperforms the state-of-the-arts with higher-quality code summaries on key metrics, including BLEU, METEOR, and ROUGE.

## CCS Concepts

• **Information systems** → *Document representation*; • **Software and its engineering** → **Software maintenance tools**.

## Keywords

source code summarization, semantic extract, abstract syntax tree, control flow graph

## 1 Introduction

Recently, technological innovation, the open-source ecosystem, and evolving user demands have driven the rapid development of software systems, which are increasingly replacing more traditional interaction scenarios and solving a broader range of practical problems. As a result, the scale and complexity of software systems have significantly increased, with developers often spending more effort on maintaining existing functions than on developing new ones [35]. There is an urgent need for a method that allows developers to quickly grasp the current software system code semantics. Generating code summaries is a viable solution that creates concise natural language descriptions for source code [21], helping developers quickly understand the software system. However, code summaries in software systems often lack readability or are completely missing, failing to serve their intended purpose. The emergence of automatic code summarization techniques has effectively alleviated these issues by generating high-quality summaries without the need for manually reading through all the code [28]. Research on code summarization can be divided into three categories based on the scope of abstraction: statement-level summaries [26], which aim to explain the meaning of individual code statements; function-level summaries [8], which aim to outline the main functions of a function; and file-level summaries [7], which aim to describe the intent of an entire code file. Our study concentrates on function-level code summarization, as it provides a balanced degree of granularity, avoiding the excessive detail of statement-level analysis while not becoming overly general like file-level summarization.

**Existing Works.** Early code summarization employed *rule and template based methods* [22]. Initially, a programming language was selected, and corresponding templates were manually customized. Summaries were generated by populating these templates according to a pre-determined set of information extraction rules. Subsequently, *information retrieval (IR) methods* [11] [12] became widely adopted, with the core approach being to measure the association between code statements through vector operations, as noted by [32]. In 2012, Hindle [15] and others proposed the "naturalness" hypothesis of code, which posits that most code statements are natural and inherently regular like natural language. Researchers then began using *deep learning-based approaches*, believing that neural network models could leverage large datasets of code to learn complex contextual features, similarly to how natural language is processed, thus generating more accurate code summaries. Research has predominantly utilized sequence-to-sequence models as the overarching framework, employing various time-series models in the encoder and decoder to process features of the source code, and incorporating models such as CNN [30], LSTM [17], GCN [5], and transformer [19]. In terms of feature extraction, most existing studies opt for the Abstract Syntax Tree (AST) as the code's feature representation, where each node in the AST represents a syntactic element, and nodes are combined in a hierarchical tree structure, capturing both the syntactic and structural information of the source code.

**Research Gaps.** However, the aforementioned methods have their shortcomings. Methods based on rules and IR primarily extract superficial semantics, and improper identifier naming can severely impact the accuracy of keyword extraction, lacking scalability. Methods based on deep learning typically focus only on the AST, but due to the sparse information within AST nodes and the high number of nodes, the information becomes fragmented. This results in an inability to provide continuous syntactic information and a comprehensive view of control dependencies to the model.

**Motivation.** To address the aforementioned issues, we propose an enhanced semantic extractor with multi-granularity feature fusion. We extract code tokens, preorder traversal sequences of AST, and Control Flow Graph (CFG) features from the source code, tokenize them using the BPE method, and embed them as feature

vectors. The code tokens are input into the CodeBERT pre-trained model to obtain context representations of the same dimension. The AST sequences are input into a decoupled attention module based on relative position encoding, allowing nodes to aggregate semantic information from their siblings, ancestors, and descendants. CFG features are initially aggregated using a GCN and then global dependencies are captured through an attention mechanism. Subsequently, the encoded features of adjacent granularities are input into a cross-attention module to achieve feature fusion. Finally, the intermediate representation of the fused features is input into the decoder module of a transformer architecture to generate the summary. By preprocessing and encoding the multi-granularity features of the source code, complete syntactic and structural information of the source code can be learned, addressing the issue of fragmented information representation in existing methods.

**Contributions.** Our contributions are four-folds, summarized as follows.

- We designed a novel AST structure generation method that compresses the number of nodes and removes redundant data, increasing the syntactic information density of individual nodes, improving the model's ability to encode AST features.
- The encoder of MGF-ESE extracts features at three different granularity levels, employing tailored encoding methods for multi-dimensional representation of the source code, effectively capturing various layers of semantic information.
- We applied a cross-attention module to fuse multi-granularity source code information from low to high levels, enhancing the model's understanding of overall contextual information.
- Experiments on selected datasets demonstrate the advanced performance of MGF-ESE in summary generation, with ablation studies confirming the robustness of each module.

## 2 Related Work

### 2.1 Code Summarization Based on Neural Network Models

With the continuous development of neural networks, researchers have explored ways to enhance the quality of code summarization through more advanced foundational models. Early models for code summarization were primarily based on RNNs for sequence modeling. Iyer et al. [17] introduced the first deep learning-based model for code summarization, employing a LSTM network as an encoder to generate vector representations of source code. This vector is then fed into another RNN decoder to generate the code summary. Allamanis et al.[2] introduced a convolutional attention mechanism to effectively detect local time-invariant and long-range topical attention features in source code. With the widespread adoption of the Transformer model in the NLP field [4], Ahmad et al. [1] began exploring the attention mechanisms within Transformers to capture the long-distance dependencies between code elements. Since then, most research in code summarization has been based on the Transformer model. Building on this, Choi et al. [5] integrated GNNs, which effectively encode the graphical structural features of code. On the other hand, Li et al. [19] leveraged CNNs to reduce data dimensions, thereby conserving computational resources. However, due to the length of source code features, capturing long-distance dependencies remains a challenge.

### 2.2 AST Processing in Code Summarization

AST are one of the most fundamental features of source code [18]. Improper processing of ASTs can lead to difficulties for models in understanding and handling the syntactic and structural information of source code. There are mainly two approaches to addressing this issue: The first approach involves using specific folding algorithms for linearization. For example, Hu et al. [16] proposed a novel Structured-Based Traversal (SBT) method for serializing ASTs. Building on this, MRNCS [10] improved the method by removing irrelevant syntactic nodes while preserving the core semantics of the AST, termed as Simplified Syntax Tree (SST). The second approach utilizes appropriate encoding techniques and neural networks to model ASTs. Shido et al. [27] introduced the Tree-LSTM model, an extension of the traditional LSTM that accepts tree-shaped data as input, allowing each node to aggregate information from its parent and child nodes. MMCS [36] enhanced this model by adding different types of edges to form a heterogeneous graph, improving the capture of implicit relationships between AST nodes. The AST-trans model [31] employs relative position encoding and decoupled attention to focus on the feature aggregation between ancestor-descendant and sibling nodes in ASTs, significantly reducing time complexity. Finally, the CSA-trans model [23] uses the SBM attention mechanism, which allows the model to learn more global node relationships within the AST and improves computational efficiency by simplifying the attention aggregation strategy. However, these methods overlook the extraction of other granular features from source code, leading to diminishing marginal utility in processing AST representations, resulting in difficulties in generating high-quality code summaries.

In summary, while recent studies have employed advanced model architectures to encode ASTs using improved serialization techniques and tree structure modeling with neural networks, they still fall short in fully capturing features across varying levels of granularity. Enhancing code summarization by incorporating multi-granularity feature extraction is thus a central goal of our ongoing work.

## 3 Preliminary and Problem Formulation

### 3.1 Preliminary

**CodeBERT Pre-Trained Model:** CodeBERT [7] adopts a multi-layer bidirectional Transformer architecture, capable of performing masked language modeling on bimodal data (Natural Language (NL) descriptions and Programming Language (PL) code). By predicting randomly masked tokens in the input sequence, the model enhances its understanding of unlabeled programming language data.

**Attention Mechanism [33]:** When encoding AST and CFG features, the vector updates between nodes are implemented through attention mechanisms. This mechanism simulates human-like attention by assigning varying levels of importance to different parts of the specific code being processed. The core of this mechanism is the scaled dot-product attention operation, which primarily involves transforming the input data through linear projections to generate corresponding Q, K, and V matrices. The attention weights are computed by calculating the dot product between the Q and K matrices, followed by weighted updates on the V matrix.

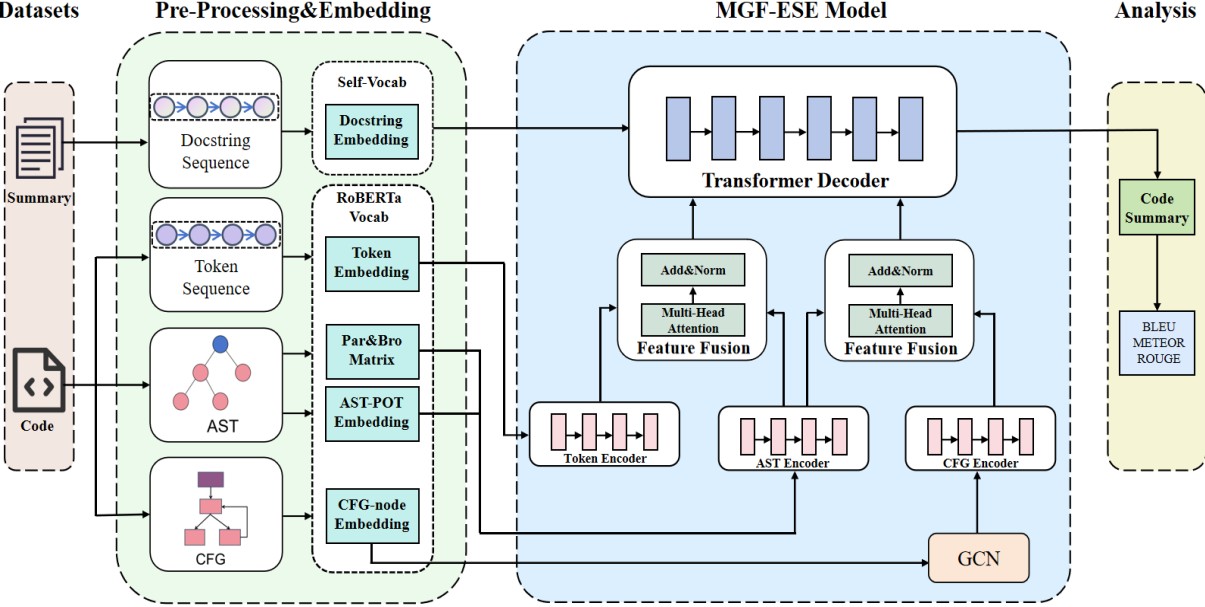

**Figure 1: Overview of MGF-ESE Model. Input code features, including code tokens, AST, and CFG, are preprocessed and encoded to capture multi-granularity semantics before being fed into the Transformer-based decoder for summarization generation.**

Compared to traditional attention mechanisms, where positional encoding (PE) is directly added to the semantic embeddings, disentangled attention [13] introduces separate positional encoding vectors for each node in addition to their content encoding vectors. When calculating attention weights, this mechanism not only computes the correlation between content vectors but also considers the impact of content-to-position and position-to-content relationships on the final results. In the feature sequence of source code, the correlation between two nodes is significantly influenced by their relative positions. Disentangled attention can effectively model the influence of both semantic and positional information on the overall correlation between nodes.

### 3.2 Problem Definition

Automatic code summarization is a key task in the field of intelligent software engineering, with significant implications for practical applications such as system maintenance and code review. The objective of code summarization is to generate a corresponding comment $N = \{n_1, \ldots, n_i\}$ for the source code $C = \{c_1, \ldots, c_i\}$. The required dataset $D$ for the experiment consists of N pairs of source code and comments (C, N). We use the training set to train the model's ability to understand the source code, the validation set to fine-tune the model, and the test set to evaluate the model's performance. The optimization function of this model is defined as follows:

$$L = -\frac{1}{N} \sum_{i=1}^{N} \sum_{t=1}^{T} \log[p(y_t|y_1, \ldots, y_{t-1})], \quad (1)$$

Essentially, cross-entropy guides the optimization of the learnable parameters in the model by quantifying the discrepancy between the predicted probability distribution of the summary words and

their actual distribution, thereby enhancing the model's understanding of the source code.

## 4 Method

### 4.1 Overview

The structure of the proposed MGF-ESE is illustrated in Figure 1 and primarily comprises four main components: a code token encoder, an AST encoder, a CFG encoder, and a decoder. The code token encoder inputs feature $F_{Tok} \in \mathbb{R}^{l_{Tok} \times d_{model}}$ into a CodeBERT pre-trained model to obtain context vectors of the same dimension and generates the Query matrix for subsequent cross-attention modules. The AST encoder takes as input the AST's preorder traversal sequence $F_{Ast} \in \mathbb{R}^{l_{Ast} \times d_{model}}$, sibling relationship matrix, and ancestor-descendant relationship matrix, acquires relative positional embeddings through the inter-node distance relationships, and finally aggregates node semantic information within a predefined strong relationship scope via a decoupled attention module. The CFG encoder inputs features in format $G(V, E)$ into GCN, aggregates features, and arranges them according to the execution order of CFG nodes in the program, feeding the resulting CFG feature matrix $F_{Cfg} \in \mathbb{R}^{l_{Cfg} \times d_{model}}$ into an attention module. The AST encoder is stacked with $N$ decoupled attention modules, while the CFG encoder is stacked with $N$ GNN modules and $N$ self-attention modules. The decoder receives intermediate vector representations generated by two cross-attention modules and consists of $N$ identical decoder layers stacked together.

### 4.2 Pre-Processing and Embedding

**Code Token Pre-processing:** In high-quality source code, function names are typically composed using PascalCase or camelCase

naming conventions to represent their overall functionality, serving as a form of summary. Given that the CodeBERT pre-trained model can accept bicameral data as input, we tokenize the function names from the source code code token and extract them as the NL component of the input sequence. In the source code, their original positions are uniformly replaced with "func name". The overall sequence is treated as the PL part.

**AST Pre-Processing:** We utilize the $ast.parse()$ method from the Python language's AST module to obtain the AST string representation of Python source code. However, the string contains a significant amount of redundancy due to nodes with null values. To address this, we have developed a graph structure generation method for the AST, which effectively filters out redundant information from the AST string representation. This method also identifies a unique parent node for all nodes other than the root node, forming a tree data structure, as shown in Algorithm 1. This algorithm divides the string representation of an AST into units based on nodes while preserving parentheses. When string elements are followed by formats such as "=value", "=number", or "=word", they are segmented into individual nodes. Finally, based on the structural information retained by the parentheses, each divided node is assigned a unique parent node, resulting in a tree structure of the AST. Ultimately, the publicly disclosed data processing method by AST-trans is used to obtain the pre-order traversal sequence of the AST and the node relationship matrix.

**CFG Pre-Processing:** The method for generating the CFG graph structure of Python source code in this article is inspired by the core ideas of the pycfg package. However, since this package is no longer maintained and its functionality has various imperfections, extensive improvements have been made based on this package. Additional recognizable node types have been introduced, and potential erroneous paths, such as those that might occur while parsing try and catch blocks, have been corrected. The control flow direction when the program catches errors has also been refined. The generated CFG graph structure consists of nodes based on statements, and concatenating all nodes can reconstruct the content of the code. The CFG of the Java source code is generated using the angr tool.

**Embedding:** To mitigate the Out-of-Vocabulary (OOV) problem, all features of the source code are tokenized using Byte Pair Encoding (BPE) [25], and each token is mapped to a unique index in the vocabulary. Subsequently, each index is used to extract a fixed-dimensional vector from the embedding matrix. For each modality, the feature sequence is set to a maximum length, sequences shorter than the specified length are padded with the $< pad >$ tokens, while longer sequences are truncated. The source code code tokens are embedded following the above steps, with $< cls >$ and $< sep >$ tokens added to the beginning and end of the index sequence the index sequence to indicate the start and end positions. For the AST, each node is tokenized, and if the tokenized length exceeds one, a max-pooling operation is applied to aggregate multiple vectors into a single vector representation. As for the CFG, since each individual node contains relatively rich semantic features, we treat the node as a code token and perform pre-training. The embedding vector of the first $< cls >$ token in the output is used to represent the overall semantics of the node. By using CodeBERT for pre-training, the model can bypass the need to train an embedding layer with a large number of parameters while also obtaining single-vector

---

**Algorithm 1** Tree Structure Representation from AST String

**Input:** $input\_string$ (String form of AST)
**Output:** $Valid\_nodes$(valid keyword in AST), and $edges$ (a list of edges representing parent-child relationships)
1: **function** TREE_STRUCTURED_AST($input\_string$)
2:     $pattern \leftarrow$ regex for words followed by '=value', '=number', '=word', or '('
3:     $matches \leftarrow pattern.finditer(input\_string)$
4:     $Structure\_node \leftarrow [], last\_end \leftarrow 0$
5:     **for** each match in matches **do**
6:         Process brackets between matches and append to $Structure\_node$
7:         Extract and append keywords or brackets based on match pattern
8:     **end for**
9:     Process remaining brackets after last match
10:     $Valid\_nodes \leftarrow \{i : Structure\_node[i] \notin' \; [](\;)'\}$
11:     $edges \leftarrow []$
12:     **for** $i$ in $range(1, length(Valid\_nodes))$ **do**
13:         $child\_index \leftarrow Valid\_nodes[i]$
14:         $parent\_index \leftarrow$ FindParentElement($Structure\_node, child\_index$)
15:         **if** $parent\_index$ is not None **then**
16:             $edges.append((parent\_index, child\_index))$
17:         **end if**
18:     **end for**
19:     **return** $Valid\_nodes, edges$
20: **end function**

---

representations for individual nodes. Due to the small number of CFG nodes and the manageable space required to store node embeddings, the choice has been made to enhance the model's training speed by pre-embedding the node features. The model encoder in this article also uses the vocabulary corresponding to the pre-trained model.

### 4.3 MGF-ESE Encoder

**Code Token Encoder:** The code token transformed from source code fragments contains rich lexical and semantic information. Processing the features of the code token provides the model with a fundamental understanding of the source code. Taking into account the balance between computational efficiency and model effectiveness, we use the CodeBERT model as the code token encoder for our model, as it achieves SOTA performance in code token representation. The encoding process using CodeBERT is as follows:

$$\tilde{H}_{Tok} = CodeBERT(E_{Tok}), \tag{2}$$

$$H_{Tok} = W_t \cdot \tilde{H}_{Tok}, \tag{3}$$

where $H_{Tok} \in \mathbb{R}^{l \times d_{model}}$ is the context vector processed by Code-BERT. The role of the fully connected layer $W_t$ is to further fine-tune the patterns and features relevant to the task at hand, based on the context vectors generated by the general pre-trained model, so that the model's representational capacity can better focus on the specific characteristics and requirements of the current dataset.

**AST Encoder:** Since the AST contains a large number of nodes representing the syntactic types of source code, it is typically longer than the original code sequence. Traditional Transformer encoders compute the correlation between each individual node and all other nodes in the AST using the attention mechanism, which incurs substantial computational overhead. For nodes with relatively large distances and weak semantic correlations, calculating the correlation scores results in higher computational costs compared to effectively aggregating the embeddings of relevant nodes. Based on the above observations and the existing work of AST-Trans, we limit the aggregation scope of nodes to their sibling and parent-child nodes, and further represent the closeness of relationships using relative distances. This relative distance is embedded as an independent relative distance vector and incorporated into the disentangled attention mechanism to encode the AST. The specific encoding process is as follows:

For the tree structure of the AST, we adopt a preorder traversal to serialize it. Although the SBT method uses parentheses to better preserve the original structure, it significantly increases the serialized sequence length, which in turn raises the complexity of capturing node relationships through matrices. Compared to inorder and postorder traversals, the advantage of preorder traversal is its ability to partially restore program statements. Although it may lead to some information loss and cannot fully recover the tree structure, it prevents excessive expansion of the sequence length and helps maintain the model's ability to extract key information effectively.

We define the $S \in \mathbb{R}^{N \times N}$ matrix and $P \in \mathbb{R}^{N \times N}$ matrix to store ancestor-descendant relationships (existence of a path from the root node to both nodes) and sibling relationships (nodes sharing a common parent) between nodes. Let $N$ be the total number of nodes. The top-down and left-right directions are defined as positive directions. If the $i$-th node in the sequence is the grandparent of the $j$-th node, then the shortest path distance between the two nodes is 2, $p_{ji} = 2$, while the corresponding $p_{ij} = -2$. To further constrain the aggregation range of nodes and reduce the spatial complexity of processing data, we set a maximum relative distance threshold $K$ for both types of relationships. If the relative distance between two nodes exceeds this threshold, they are considered to have no relationship, and the corresponding position in the matrix is set to infinity. The specific rules for matrix definitions are as follows:

$$p_{ij} = \begin{cases} \mathrm{PAR}(i,j) & \text{if } |\mathrm{PAR}(i,j)| \leq K, \\ \infty & \text{if } |\mathrm{PAR}(i,j)| > K. \end{cases}$$

$$s_{ij} = \begin{cases} \mathrm{SIB}(i,j) & \text{if } |\mathrm{SIB}(i,j)| \leq K, \\ \infty & \text{if } |\mathrm{SIB}(i,j)| > K. \end{cases} \tag{4}$$

Next, we transform the inter-node relationships stored in the matrices into tree-structure-based relative position embeddings. Since any given pair of nodes can only have either a ancestor-descendant or sibling relationship, we use $r_{ji}$ to uniformly represent $s_{ji}$ or $p_{ji}$ and derive a unique relative distance index based on the value of $r_{ji}$. The definition rules of $\delta_r(i,j)$ are as follows, values

greater than zero are considered strong relationship nodes of $i$:

$$\delta(i,j) = \begin{cases} r_{ij} + K + 1 & \text{if } r_{ij} \in [-K, K], \\ 0 & \text{if } r_{ij} = \infty. \end{cases} \tag{5}$$

As previously mentioned, we define a positive direction for the relative relationships between nodes to distinguish the hierarchical relationships for the same pair of nodes. Therefore, positive and negative indices with the same absolute value are embedded differently. Since the relative position embedding matrix does not have separate columns for positive and negative indices, an offset value of $P + 1$ is applied. Consequently, when calculating the correlation of node $i$ to node $j$, the process is as follows:

$$\alpha_{i,j} = \mathbf{Q}(x_i)\mathbf{K}(x_j)^\top + \mathbf{Q}(x_i){\mathbf{K}^P_{\delta(i,j)}}^\top + \mathbf{Q}^P_{\delta(j,i)}\mathbf{K}(x_j)^\top. \tag{6}$$

In the above equation, $x_i$ and $x_j$ represent the embeddings of the $i$-th and $j$-th nodes in the preorder traversal sequence respectively. We define a content-based query function $Q$ and key function $K$, along with a relative position-based query function $Q^P$ and key function $K^P$. $Q^P_{\delta(j,i)}$ denotes the $\delta(j,i)$-th row of $Q^P$, and $K^P_{\delta(i,j)}$ denotes the $\delta(i,j)$-th of $K^P$. In this manner, we transform the relative position indices into independent relative position vectors and assess the impact of relative positions on the final correlation between token pairs through additional content-to-position and position-to-content computations.

Based on the correlation scores between nodes and their strongly related nodes, we utilize disentangled multi-head attention to update the original feature, where each head only considers either the ancestor-descendant or sibling relationships. The outputs are then concatenated. Since three terms are computed in Equation (4), the corresponding scaling factor is adjusted to $\frac{1}{\sqrt{3d}}$. Accordingly, we define a content-based value function $V$ and a relative distance-based value function $V^P$. During the final node feature update, we only focus on nodes with $\delta(i,j) > 0$.

$$o_i = \sum_{j}^{j \in \{j \mid \delta(i,j) > 0\}} \sigma\left(\frac{\alpha_{i,j}}{\sqrt{3d}}\right)\left(V(x_j) + V^P_{R_{ij}}\right). \tag{7}$$

**CFG Encoder:** Compared to the AST, the number of nodes in the CFG is significantly reduced, and applying a disentangled attention mechanism does not effectively decrease the number of nodes being processed. For non-branch and non-leaf nodes, the in-degrees and out-degrees are very limited, typically having only one outgoing edge and one incoming edge. For branch nodes, such as those in loops, conditionals, and exception handling structures, there are more edges connected. Based on these characteristics, we first apply a Graph Convolutional Network (GCN) to perform initial feature aggregation on the embedding matrix of the CFG. The graph convolution performs message passing between neighboring nodes based on the normalization of node degrees. Since a single CFG node contains relatively rich semantic information, the GCN allows non-branch nodes to aggregate the semantic features of surrounding branch nodes without causing feature homogenization due to high aggregation levels. The processing steps of GCN are shown as follows:

$$H^{(l+1)} = \sigma(\tilde{D}^{-\frac{1}{2}}\tilde{A}\tilde{D}^{-\frac{1}{2}}H^{(l)}W^{(l)}), \tag{8}$$

$$h_i^{(l+1)} = \sigma \left( \sum_{j \in N(i) \cup \{i\}} \frac{1}{\sqrt{d_i d_j}} W^{(l)} h_j^{(l)} \right). \tag{9}$$

where $H^{(l)}$ represents the feature matrix of CFG nodes at layer $l$, and $H^{(l+1)}$ is the updated node feature matrix after the operation.$\tilde{A}$ is the adjacency matrix containing connectivity information of nodes within the CFG, augmented with an identity matrix to ensure self-features are aggregated during updates.$\tilde{D}$ is the degree matrix of $\tilde{A}$, with diagonal elements representing the sum of each row of $\tilde{A}$,including the node's out-degree, in-degree, and itself. $\tilde{D}^{-\frac{1}{2}}$ represents the inverse square root of the degree matrix and $\tilde{D}^{-\frac{1}{2}} \tilde{A} \tilde{D}^{-\frac{1}{2}}$ performs normalization based on node degrees to control the scale of feature propagation across different nodes, preventing excessively high-degree nodes from dominating feature updates, which could lead to either explosion or vanishing of features during the update process. Equation 7 specifies the update formula for an individual node.$\sqrt{d_i d_j}$ implement specific normalization effects where higher-degree adjacent nodes contribute less weight to the node's feature integration.$W^{(l)}$denotes the learnable weight matrix that transforms node feature representations at each layer, followed by the introduction of a non-linear activation function.

Due to the high semantic information density within CFG nodes and the potential strong relationships between nodes that are far apart (implicit control and data dependency paths), after an initial aggregation of node features through GCN layers, we construct a multi-head self-attention module based on the standard transformer architecture. This module is used to further compute the intermediate representation of CFG features, capturing the global dependencies within the sequence. The process is illustrated as follows:

$$H_{cfg} = Softmax \left( \frac{Q(x_c)K(x_c)^T}{\sqrt{d}} \right) V(x_c). \tag{10}$$

In the above formula, $x_c$ represents the feature matrix of the CFG. $Q$, $K$ and $V$ respectively stand for the query function, key function, and value function. By aggregating features both locally and globally, we obtain the final contextual representation of the CFG node sequence.

## 4.4 Feature Fusion

In the previous steps, we independently processed three distinct levels of granularity features. Next, we utilize a cross-attention module to fuse the intermediate representations of these features. Following the principle of integrating high-granularity information into low-granularity features, the specific processing steps are as follows:

$$H_{cross\_Token} = Softmax \left( \frac{Q(H_{Tok})K(H_{Ast})^T}{\sqrt{d}} \right) V(H_{Ast}). \tag{11}$$

$$H_{cross\_AST} = Softmax \left( \frac{Q(H_{Ast})K(H_{Cfg})^T}{\sqrt{d}} \right) V(H_{Cfg}). \tag{12}$$

## 4.5 MGF-ESE Decoder

The decoder used in this paper is based on the traditional Transformer architecture's decoder section, designed for generating code summarization. The decoder is composed of $N$ stacked decoder layers, each divided into three parts. The first part includes a masked multi-head self-attention mechanism, residual connections, and normalization. The masking mechanism ensures that the model relies only on the information already output at the current time step. The second part comprises a cross-attention module, residual connections, and normalization, where The cross-attention module is designed to integrate features from code tokens with AST and from AST with CFG. These features are combined with the output from the previous part. The third part includes a feed-forward network, residual connections, and normalization, which further capture deeper features. Finally, a linear layer of dimension $d_{\text{model}} \times d_{\text{vocab}}$ and a softmax activation function are used to calculate the probability of generating each word in the vocabulary at the current time step, with the highest probability index corresponding to the output word for that time step.

## 5 Experiments

## 5.1 Experiment Setup

**Dataset:** This paper evaluates the effectiveness of the model using datasets in two programming languages. The Java dataset is adopted from the dataset used in the DeepCom method proposed by Hu et al. [16], which was collected from high-quality open-source code on GitHub during 2015-2016 and is considered a classic dataset in code summarization research. The Python dataset is sourced from the dataset provided by Wan et al [34]. This study focuses on generating source code summaries at the function level. During data preprocessing, bimodal data is converted into unimodal data, comments in the code are removed, and samples with syntax errors are filtered out. The selected datasets are shuffled and split into train, valid, and test sets in a ratio of 8:1:1.

**Evaluation Metrics:** In this paper, we select widely used evaluation metrics for code summarization research, including BLEU, METEOR, and ROUGE-L, detailed as follows::

- **BLEU** (Bilingual Evaluation Understudy) [24] is an accuracy-based metric that measures n-gram precision between the generated summaries and the reference labels by calculating the overlap rate of n-grams and applying a brevity penalty to penalize short translation hypotheses.
- **METEOR** (Metric for Evaluation of Translation with Explicit) [3] considers both precision $P$ and recall $R$. It evaluates the alignment between generated code summaries and reference summaries while taking into account synonyms, stemming, and paraphrasing.
- **ROUGE-L** [20] leverages the Longest Common Subsequence for evaluating the quality of summarization.

**Baseline:** The proposed model was compared with seven baseline models, all of which adopted the sequence-to-sequence architecture. The detailed information is summarized as follows:

- CODE-NN [17] uses an attention-based LSTM network to generate summaries.

**Table 1: Comparison of MGF-ESE with the baseline methods. Our method MGF-ESE demonstrates the best performance on all datasets compared with other baseline models.**

| Methods | Input | Java | | | Python | | |
|---|---|---|---|---|---|---|---|
| | | BLEU-4 | ROUGE-L | METEOR | BLEU-4 | ROUGE-L | METEOR |
| CODE-NN [17] | Code | 27.60 | 41.10 | 12.61 | 17.36 | 37.81 | 09.29 |
| Dual Model [35] | Code | 42.39 | 53.61 | 25.77 | 21.80 | 39.45 | 11.14 |
| Tree2Seq [6] | AST | 37.88 | 51.50 | 22.55 | 20.07 | 35.64 | 08.96 |
| Transformer+GNN [5] | AST | 45.49 | 54.82 | 27.17 | 32.82 | 46.81 | 20.12 |
| AST-Trans [31] | AST | 45.60 | 55.27 | 28.65 | 34.27 | 47.02 | 20.19 |
| CSA-Trans [23] | AST | 45.95 | 56.02 | 29.24 | 35.69 | 49.05 | 21.13 |
| DeepCom [16] | Code+AST | 39.75 | 52.67 | 23.06 | 20.78 | 37.35 | 09.98 |
| GREAT [14] | Code+AST | 44.97 | 54.42 | 27.15 | 32.11 | 46.01 | 19.75 |
| MGF-ESE | Code+AST+CFG | **46.79** | **56.48** | **29.93** | **36.37** | **50.22** | **21.84** |

- Dual Model [35] performs both code summarization and generation tasks simultaneously through dual learning
- Tree2Seq [6] uses tree-based LSTM as the encoder to capture the structural information of AST
- Transformer + GNN [5] applies a transformer-based decoder to process the AST encoded by the GNN.
- AST-Trans [31] aggregates node features with only two types of relative relationships when applying disentangled attention.
- CSA-Trans [23] uses stochastic block model (SBM) attention for improved node relationship extraction
- DeepCom [16] first applies the SBT to linearize the AST.
- GREAT [14] leverages code tokens enriched with diverse relational cues derived from the AST.

Because we are using the same dataset as SG-Trans [9], the performance of our six baseline models comes from the literature. We have reproduced the best-performing baseline model, CSA-Trans. For AST-Trans, we employed the open-source method by Sun et al. [29] to process the source code into an inputable format and trained it using the hyperparameters provided in the paper.
**Implement Details:** The basic experimental settings are as follows: the embedding dimension $d$ for a single token is set to 768. In the AST encoder, the maximum relative distances for ancestor-descendant and sibling relationships are set to 10 and 5, respectively. The AST and CFG encoders, along with the model decoder, are stacked in four layers. The dimension of the feedforward layer is 2048. We set the batch size to 32 and use the Adam optimizer for weight updates, with a learning rate of 0.001. We employ dropout strategy and early stopping mechanism during training to prevent overfitting, with a dropout probability of 0.9 and a patience setting of 15.

## 5.2 The Effectiveness of Our Method

To thoroughly evaluate the performance of the MGF-ESE model in the task of code summarization generation, we conduct a series of comparative experiments against eight baseline models.

Table 1 displays a comparison of MGF-ESE with eight baseline models on key metrics. Due to Java having more explicit definitions of syntax and structure that facilitate feature extraction, any model performs significantly better on Java code summarization than on Python code. Among all models, MGF-ESE stands out, achieving the highest scores on the key metrics. Compared to code token-based models such as CODE-NN and Dual Model, MGF-ESE demonstrates substantial improvements; for example, it exceeds Dual Model by 10.37%, 16.14%, and 5.35% in BLEU-4, Meteor, and Rouge-L scores respectively. code tokens only reflect the most basic semantic features and do not fully express the structural characteristics of the source code. In models based on code structural features, MGF-ESE has a clear advantage over the traditional SBT serialization of AST used by DeepCom, achieving higher scores of 17.71%, 29.79%, and 7.23% in BLEU-4, Meteor, and Rouge-L. SBT serialization significantly increases the length of the AST sequence, which is detrimental to the model's ability to extract key information from the sequences. Compared to models that model AST in a tree-based structure such as AST-Trans and CSA-Trans, MGF-ESE also shows significant improvements. Compared to the state-of-the-art model CSA-Trans, improvements of 1.82%, 2.35%, and 0.82% in BLEU-4, METEOR, and ROUGE-L metrics on the Java dataset and 1.90%, 3.36%, and 2.72% on the Python dataset are achieved. Overall, MGF-ESE enhances the quality of summary generation by extracting multi-granularity features of the source code. code tokens and CFG are extracted from the source code to overcome the limitations of AST in expressing semantic and structural information. Additionally, a cross-attention module is employed to further integrate code tokens and CFG features, thus enabling the model to understand deep associations between features across different granularities.

## 5.3 Ablation Study

We perform an ablation study to evaluate the effectiveness of individual components of the MGF-ESE model. "w/o Token" refers to not using the cross-attention module and excluding token feature

**Table 2: Ablation study on each component of MGF-ESE for Java and Python datasets**

| Dataset | Component | BLEU-4 | METEOR | ROUGE-L |
|---------|-----------|--------|--------|---------|
| Java | w/o Token | 45.95 | 28.95 | 55.11 |
| | w/o CFG | 46.02 | 29.11 | 55.38 |
| | w/o AST | 45.29 | 28.69 | 54.92 |
| | MGF-ESE | 46.79 | 29.93 | 56.48 |
| Python | w/o Token | 34.88 | 20.97 | 48.46 |
| | w/o CFG | 35.45 | 21.20 | 49.01 |
| | w/o AST | 34.59 | 20.45 | 48.10 |
| | MGF-ESE | 36.37 | 21.84 | 50.22 |

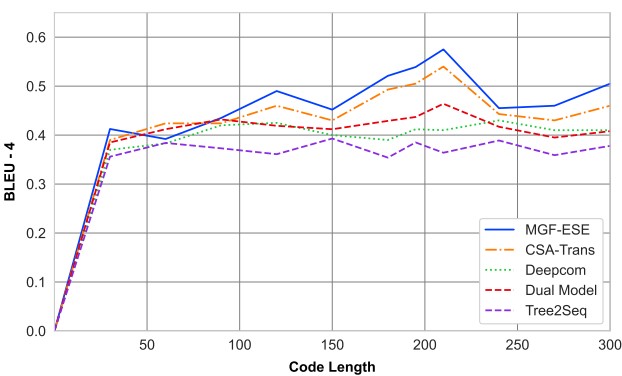

**Figure 2: BLEU-4 of Baseline Model and MGF-ESE at Different Code Lengths**

extraction, while "w/o CFG" refers to not using the cross-attention module and excluding CFG feature extraction. "w/o AST" refers to excluding AST feature extraction. The experimental results are shown in Table 2.

Among the three modules, removing the AST encoder causes the most significant degradation in model performance. In the Java dataset, BLEU-4, METEOR, and ROUGE-L scores decrease by 3.22%, 4.14%, and 2.76%, respectively, while in the Python dataset, they drop by 4.89%, 6.36%, and 4.22%, respectively. These results suggest that without AST feature extraction, relying on code tokens with significant granularity differences and CFG is detrimental to capturing the overall semantics of the source code. This also demonstrates that AST, which integrates both syntactic and structural information, is the most important feature in source code analysis and assists the decoder in computing attention over the fused features of code tokens and CFG. Furthermore, the model performs better when removing the CFG encoder than when removing the code token encoder, indicating that when AST provides syntactic and structural information, extracting additional basic syntactic information from code tokens helps better capture the overall semantics of the source code. It is also worth noting that in all ablation experiments, the performance decline in the Python dataset is greater than in the Java dataset, suggesting that the model is more sensitive to the loss of semantic features in Python.

In conclusion, although the contributions of different modules in MGF-ESE vary, each module enhances the overall performance of the model.

## 5.4 The Impact of Code Length on Model Performance

Figure 2 provides a detailed examination of how code length affects the performance of five models on the BLEU metric within a Java dataset. The analysis indicates that all models exhibit poor performance when the code length is approximately 50. This is primarily due to the insufficient information content in shorter code segments, which hampers the models' ability to extract essential information effectively, thus impacting their performance. As the code length increases to between 50 and 200, the models show a fluctuating improvement in performance. Within this range, the MGF-ESE model is particularly notable, benefiting from its efficient strategy of node aggregation in the AST encoder and its capability to extract multi-granularity features, thereby surpassing all baseline models in performance. However, when the code length exceeds

200, all models experience a decline in performance due to the increased difficulty in capturing long-distance dependencies and rising computational complexity. Nevertheless, the CSA-Trans and MGF-ESE models, which are based on the Transformer architecture, exhibit less performance degradation due to their effective handling of these long-distance dependencies. Notably, the MGF-ESE model, with its advantage in capturing and further fusing multi-granularity features of the source code, outperforms the CSA-Trans model in overall performance.

## 5.5 Limitation

The experiments further confirm the complexity of balancing feature extraction granularity across different programming languages. The model is particularly sensitive to the loss of semantic features in Python, as evidenced by the significant performance decline observed in the ablation studies. This underscores the need for more precise feature integration strategies that can accommodate the structural and semantic differences between languages.

## 6 Conclusion

In this paper, we introduce an Enhanced Semantic Extractor with Multi-Granularity Feature Fusion (MGF-ESE). This model effectively processes features of the AST by extending the semantic information of individual nodes and aggregating features of highly related nodes, while also reducing computational overhead. To compensate for potential semantic gaps arising solely from AST feature extraction, we additionally extract code tokens and CFG, integrating these with the AST through a cross-attention module. Furthermore, ablation studies conducted on the encoder confirm that each module significantly contributes to the overall performance of the model. Our analysis of code length further substantiates that capturing high-granularity features of the source code can significantly enhance the quality of generated summaries for lengthy code. In the future, we plan to continue this line of research by exploring more effective methods for integrating features of varying granularities and attempting to apply our model to code summarization tasks in other programming languages through transfer learning.

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
