# OpenReview forum: "MGF-ESE: An Enhanced Semantic Extractor with Multi-Granularity Feature Fusion for Code Summarization"
_ACM.org/TheWebConf/2025/Conference — WWW 2025 Poster_

### Official Review · Reviewer_2d5z · 2024-11-04

**Novelty:** 4
**Technical Quality:** 4

**Review:**

Paper Summary:
In this paper, the authors proposed a multi-granularity feature fusion framework for code summarization tasks. Specifically, MGF-ESE comprises four key components a CFG embedding, an AST embedding, a token sequence embedding, a docstring sequence embedding, and a traditional transformer. Compared to existing methods, the experimental results are relatively ideal

Summary Of Strengths:
1. Compared to existing methods, the experimental results are relatively ideal.

Summary Of Weaknesses:
1. The foundational model is anchored in the BERT architecture, yet it remains untouched by the advancements of contemporary large language models (LLMs).
2. The absence of open-source code.
3. The approach merely introduces an additional embedding layer, devoid of any groundbreaking innovation.

**Questions:**

1. The AST sequences are input into a decoupled attention module based on relative position encoding, allowing nodes to aggregate semantic information from their siblings, ancestors, and descendants.  Can you explain how this mechanism disentangles different aspects of the code and how it improves over traditional attention mechanisms?

2. Integration of CFG and Tokens: Integrating CFG and code tokens with AST through cross-attention modules seems innovative.  How do you ensure that the fusion of these different representations does not lead to information overload or dilution of important signals?

3.How does the LLM excel in this particular endeavor?

**Reviewer Confidence:**

4: The reviewer is certain that the evaluation is correct and very familiar with the relevant literature

**Scope:**

3: The work is somewhat relevant to the Web and to the track, and is of narrow interest to a sub-community

---

### Official Review · Reviewer_5o99 · 2024-11-20

**Novelty:** 5
**Technical Quality:** 4

**Review:**

This paper proposes a code summarization model named MGF-ESE. Comparing with existing works, its key novelty lies in the use of the CFG features for code input modelling, while a cross-attention module is applied to fuse the different input features. Experimental results on two benchmark datasets show the effectiveness of MGF-ESE.

Strength:

S1. The use of CFG features is shown to effectively improve the quality of the output summary, which could intrigue more studies using such features.

S2. The proposed model is shown to be effective on both Python and Java datasets, showing its applicability.

Weakness:

The paper can use a thorough proofread to polish the writing and clarify some of the technical details:

The novelty of the proposed model in how it exploit the AST features compared with that by the literature should be further clarified.

It was stated in Section 4.2 that "We utilize the ast.parse() method from the Python language's AST module to obtain the AST string representation of Python source code". How about AST extraction for the Java dataset?

More details should be given for the  CFG graph generation procedure, and ideally the source code should be shared for reproducibility.

What are $t$, $T$, and $y$ in Equation 1? It would be good to show an example input and output of the problem studied (e.g., from one of the experimental datasets).

How are $H_{cross\_Token}$ and $H_{cross\_AST}$ combined together to form the input of the MGF-ESE Decoder?

Some of the contents (e.g., first sentence of the Introduction) can be simplified or removed to make room for more details of the technical contents, e.g., an example and figure can be added to illustrate the issues of existing works that are addressed in this paper, and why the proposed solution is novel.

Terms like "BPE" and MGF-ESE should be written in full the first time when they are mentioned.

Using $N$ to denote both a comment and the number of comment-code pairs (and later the number of attention/GNN modules) is odd.

Typos and grammar: "a novel AST generation method with compresses the scale of nodes" => "a novel AST generation method which compresses the scale of nodes"; "similarly to how natural language is processed" => "similar to how natural language is processed"

The authors may also want to discuss the relevance of the submission to the conference and the submitted track.

**Questions:**

See Weakness above.

**Reviewer Confidence:**

2: The reviewer is willing to defend the evaluation, but it is likely that the reviewer did not understand parts of the paper

**Scope:**

3: The work is somewhat relevant to the Web and to the track, and is of narrow interest to a sub-community

---

### Official Review · Reviewer_eD7i · 2024-11-28

**Novelty:** 6
**Technical Quality:** 6

**Review:**

The MGF-ESE framework generates code summaries by integrating multi-granularity features from source code, such as Abstract Syntax Trees (AST), Control Flow Graphs (CFG), and code tokens. It employs a novel tree structure generation method and a disentangled attention mechanism for encoding AST features. Deep associations between different features are captured using a cross-attention mechanism. These processes highlight the framework's improved precision and efficiency in producing high-quality code summaries.

(strength)

1. This paper addresses an issue of pressing need at the intersection of artificial intelligence and software engineering, advancing the application of neural networks in programming languages and providing a novel approach to code summarization research.
2. The introduction is logically structured, clearly outlining the background, significance, and limitations of existing methods in code summarization. The transitions between topics are smooth, making the content easy to read, and the concepts mentioned in the paper are accessible to readers.
3. The authors demonstrate a solid understanding of the foundational models and modules used in the proposed model, drawing inspiration from related works, which provides a robust basis for the overall model development.

(weakness)

1. In the description of the model encoder, the paper provides a detailed explanation of the foundational model's workflow. However, the extensive exposition leads to some redundancy, while there is insufficient explanation of the rationale behind using the current approach for handling features of different granularities.

2. In the limitations section, the paper focuses on a common issue in code summarization research, but lacks exploration and discussion of specific issues unique to this study. This weakens the readability and relevance of the section, making it less targeted.

3. The model's processing of multi-granularity features inevitably increases its time and space complexity. Is this cost increase reasonable, and does it yield sufficient performance improvements to justify the added complexity?

**Questions:**

See the above comments.

**Reviewer Confidence:**

4: The reviewer is certain that the evaluation is correct and very familiar with the relevant literature

**Scope:**

4: The work is relevant to the Web and to the track, and is of broad interest to the community

---

### Official Review · Reviewer_gByV · 2024-12-02

**Novelty:** 5
**Technical Quality:** 5

**Review:**

The paper proposes a code summarization approach that takes into account the code tokens, the Abstract Syntax Tree (AST), and the Control Flow Graph (CFG). These features are used to capture multi-granularity semantics and are fed into a transformer-based decoder for code summarization. The conducted evaluation showcases that the proposed approach offers state-of-the-art performance in two datasets (Python and Java) using the BLEU-4, ROUGE-L and METEOR metrics. An ablation study that showcases the importance of the AST features along with a study regarding the impact of code length on the various approaches is also provided.

The paper is well-written and the proposed architecture is described thoroughly. The idea of extracting features at three different granularity levels and fusing them using crossing attention is interesting.

The authors should provide a more detailed description of the AST serialization approach, SBT Structure-based Traversal (SBT) and the MGF-ESE AST serialization. An explicit comparison evaluation should also be provided.

The description of the experiment setup contains an error regarding the number of baselines (the authors claim 6 and 7 baselines, while they use 8).

SG-Trans is mentioned but not provided in the baselines for comparison reasons. This specific work conducted also a human-based evaluation which would be beneficial for code summarization evaluation.

A major drawback of the paper is that no code is given for reproducibility reasons.

Typos:

In abstract: a novel AST generation method with -> which

P.5 second paragraph: an offset value ofP + 1 -> of P + 1 (missing space)

**Questions:**

The authors are asked to provide explanations on the comments made in the main review section.
Most importantly, the authors must ensure reproducibility of the results

**Reviewer Confidence:**

3: The reviewer is confident but not certain that the evaluation is correct

**Scope:**

3: The work is somewhat relevant to the Web and to the track, and is of narrow interest to a sub-community

---

### Official Review · Reviewer_L11h · 2024-12-02

**Novelty:** 5
**Technical Quality:** 5

**Review:**

This paper presents a generative AI system for generating natural language summarization of functions written in a programming language. One originality of the approach compared to existing work is the use of the Control Flow Graph (CFG) in addition to the program's source code and its Abstract Syntax Tree (AST).

NOTE 1: the contribution of this paper is outside my field of expertise, hence my low confidence score below.
NOTE 2: while the problem tackled by the authors is an important and interesting one, I do not see how it relates to the web (beyond the point programming, as most tasks in the digital world nowadays, is often performed on the web, but that is not relevant). So regardless of its technical quality, I wonder whether this paper should be rejected as out-of-scope.

OTHER REMARKS:
* p2, sec 3.1 : "attention mechanisms. This mechanism" → inconsistency between plural and singular
* p3: equation (1): the index `i` of the outer sum is not used anywhere else in the equation
* p4: "We utilize the `ast.parse()` method from the Python language's AST module to obtain the AST string" → the `ast.parse()` function does not return a string, but a structured object. This makes the rest of this paragraph, as well as Algorithm 1 (on the same page) quite hard to follow.
* p8: "the code length is approximately 50" → it would be better to provide a unit (characters? lines of code? symbols?)

**Questions:**

* How do you consider this work to be in scope for The Web Conference?

* You are using a dataset from 2015-2016 to train your system on Java code. In the meantime, the Java language has significantly evolved. In particular, the compiler is now able to perform type inference, allowing the programmers to omit much information. Do you expect a change in performance with more recent Java code, as you explain the difference in performance between Java and Python by Java being more explicit?

* Conversely, in the past year, Python annotations have become more and more common, as a way to provide hints (to linters and to other programmers) about the type of objects. Did your training set include such annotations? If not, do you expect that training on more recent code would improve performance?

**Reviewer Confidence:**

1: The reviewer's evaluation is an educated guess

**Scope:**

2: The connection to the Web is incidental, e.g., use of Web data or API